# Impact of the Universal Two-Child Policy on the Workload of Community-Based Basic Public Health Services in Zhejiang Province, China

**DOI:** 10.3390/ijerph16162880

**Published:** 2019-08-12

**Authors:** Yanrong Zhao, Junfen Lin, Xiaopeng Shang, Qing Yang, Wei Wang, Yinwei Qiu

**Affiliations:** Zhejiang Provincial Center for Disease Control and Prevention, Hangzhou 310051, China

**Keywords:** basic public health services, workload, universal two-child policy

## Abstract

We aimed to quantitatively estimate the amount of pressure that was placed on basic public health care services (BPHS) due to the universal two-child policy issued in 2015 by comparing the workload change in maternal and child health management and the immunization of children. BPHS performance surveillance data from 2014 to 2018 in Zhejiang Province, China were analyzed to calculate the workload of the above three services using the equivalent method of BPHS cost estimation of community health services. From 2014 to 2018, the numbers of births from the Statistical Yearbook in Zhejiang Province were 578,000, 581,000, 624,000, 670,000, and 628,000, respectively, and those from the surveillance data were 416,941, 41,490, 434,163, 546,816, and 45,964, respectively. The number of births reached a peak in 2017, with the yearbook and surveillance data showing increases of 15.92% and 31.15%, respectively, over 2014. The workload of maternal and child health management and children’s immunization also peaked in 2017, increasing by 30.37%, 12.70%, and 4.33% over 2014, respectively. In 2018, the workload of maternal and child health management and children’s immunization dropped to 107.34%, 107.73%, and 98.81% over 2014, respectively. The indicators of maternal and child health management and children’s immunization services remained stable, and the related services did not decline, even in 2017. The maternal health management workload was more affected by the universal two-child policy than child health management and children’s immunization.

## 1. Introduction

The negatives of the one-child policy introduced in 1979 by the Chinese government have appeared gradually and include an aging population, a skewed sex ratio, and a decline in the working-age population, which could threaten economic growth. In October 2015, China announced that the one-child policy had been replaced by a universal two-child policy, allowing all couples to give birth to two children [1,2,3]. Researchers predict that the universal two-child policy transition will have many benefits [1] such as a larger population size and workforce, which will help mitigate the serious aging population situation, higher consumption, and job opportunities associated with childbearing, which will contribute to economic growth. A substantial reduction in sex-selective abortions will be seen in the future, but a reduction in the sex ratio at birth is unlikely to completely normalize for many years. The undesirable trend of increasing numbers of childless families will substantially decline under the universal two-child policy due to a reduction in the bribery of officials for permission to have more than one child and a merging of family planning organizations into the National Health and Family Planning Commission [4], all of which will help to create a more harmonious society.

However, many questions remain. Although the implementation of the universal two-child policy has made numerous families reshape their decision making in terms of the mode of delivery, with a lower cesarean delivery rate for women who intend on having a second child [5,6,7], the rate of repeated cesarean delivery for women who have a cesarean delivery history is much higher [8]. Increased medical risks of pregnancy-related complications among older women have also been reported [9]. A major concern is the limited and uneven health care resources across the country, where any increase in the birth rate resulting from the universal two-child policy will exacerbate pressure on an already stressed system.

At the initial implementation stage of the policy, many studies have focused on the allocation shortage of pediatrician and maternal beds, especially in large-scale maternal and child care hospitals [10,11,12]. Few studies have analyzed the impact on community health services (CHSs). However, for a long period, community health service institutions including community health care centers, township hospitals, and village clinics have been designed to deliver comprehensive primary health care (PHC) services such as maternal and child health care and the immunization of children. In particular, since 2009, the Chinese government has launched health care reform for the equalization of basic public health services (BPHSs) [13,14]. A national essential public health services package was issued that included the establishment of health records, health education, immunization, child health management, maternal health management, geriatric health, hypertension and type 2 diabetes management, severe mental illness management, and the surveillance and control of infectious diseases and public health emergencies [14], which is continually being expanded. The CHSs, as the health gate keepers in a hierarchical medical system, are responsible for providing the BPHS package [13,14].

From 2014 to 2018, China’s annual number of births was 16.87, 16.55, 17.86, 17.23, and 15.23 million, respectively. The number of births was higher in 2016 and 2017, while lower in 2018 when compared to 2014 and 2015 [15,16]. The dramatic change in birth rate directly affects the implementation of maternal-and child-related health services (maternal and child health management and children’s immunization). With the insufficient allocation of health service resources, it is unknown as to which service was the most affected by this one shot birth boom. Will it lead to a decline in the performance of related services? Was the increase in service workload proportional to the birth increase? These questions have yet to be answered. Although when answering reporters’ questions on the two-child policy, an officer of the National Health and Family Planning Commission said that the government was confident that the implementation of this two-child policy would not place considerable pressure on the public services of health care [17].

We aimed to quantitatively estimate how much pressure was placed on the BPHS due to the universal two-child policy by analyzing the BPHS surveillance data from 2014 to 2018 in Zhejiang Province, China.

## 2. Materials and Methods

### 2.1. Data Source and Collection

The birth data were obtained from the Zhejiang Statistical Yearbook and compared with the birth data from the BPHS surveillance, which was also the source of the maternal and child health management data. The surveillance system was set up in accordance with the Basic Public Health Service Standards of Zhejiang Province (4th Edition); BPHS progress data were reported by CHS institutions with integrated verification. The data of children’s immunization were released by the provincial center for disease control and prevention (CDC) [18,19,20,21,22].

### 2.2. Measures and Variables

The workload measurement was derived from the equivalent method of the BPHS cost estimation of the CHS institutions. In the equivalent method [23], the workload indicator of a standard clinic visit (a family physician consulting with one patient for 15 min) was defined as one equivalent value (EV). Equivalent values of all of the other PHC services (BPHS included) were then calculated as their workload indicators when compared with the standard clinic visit. For example, the workload indicator of one home visit was 60.00 person–time, so its EV was 4. The EVs of the BPHS services in Zhejiang Province were modified based on the results of Beijing-based research [23] and the related values are shown in Table 1. The volume of each service was multiplied by their EV, and then added together to produce the total workload of maternal and child health management and children’s immunization services. The workload of one service = ∑EV of the sub-item service multiplied by the volume of sub-item service.

#### 2.2.1. Workload Measurement of Maternal Health Management

The surveillance data were limited to the registration number in early pregnancy, the number of prenatal examinations, and the number of postpartum visits. To maintain a high consistency between the postpartum visited population and the 42-day postpartum examination population, the number of 42-day postpartum examinations was substituted by the number of postpartum visits. Due to the lack of related data, the workload of high-risk pregnancy management was not considered in this paper. The following formula was used to obtain the workload of maternal health management:

Workload of maternal health management = registration number in early pregnancy × 8/person + the number of prenatal examinations × 4 times × 1.5/person–time + the number of postpartum visits × (3 person–time + 1 person–time).

#### 2.2.2. Workload Measurement of Child Health Management

The surveillance data were limited to the home-visited newborn infants and health-managed children aged 0 to 6 years. For the newborn infants, the accepted home visit services were usually covered by the establishment of the health records and follow-up health management; the number of health records established for the newborn infants and the number of health-managed children aged 0 were both substituted by the number of newborn infants with home visits. The number of health managed children aged 1–2 years was estimated using the number of newborn infants with home visits in the last two years. The number of health managed children aged 3–6 years was then calculated by the health-managed children aged 0–6 years minus the estimated number of health managed children aged 0 and 1–2 years. Due to the lack of related data, the workload of high-risk infants and children with nutritional diseases was not considered. The following formula was used to obtain the workload of maternal health management:

Workload of maternal health management = the number of newborn infants with home visits × (3/person + 0.5/person) + (the number of newborn infants with home visit × 4 times + the number of newborn infants with home visits in last the two years × 2 times) × 2.5/person–time + (the number of health managed children aged 0 to 6 years − the number of newborn infants with home visit − the number of newborn infants with home visit in last two years) × 2/person.

#### 2.2.3. Workload Measurement of Children’s Immunization

There are 12 national immunization program vaccines totaling 22 doses provided for children aged 0 to 6 that are free of charge, 14 doses for primary vaccination (mainly targeting newborn infants), and eight doses for booster vaccination. The immunization information system data provided the numbers of the target children and immunized children of each vaccine included in the National Immunization Program in China. As each child must present their immunization certificate before vaccination, the number of new vaccination certificates in a certain year was substituted by the highest dose of vaccine of primary vaccination in the same year. Therefore, the workload of children’s immunization = the highest dose of vaccines in primary vaccination × 0.5/person + the total doses of vaccines included in the National Immunization Program × 1.5/dose.

## 3. Results

### 3.1. Change in the Birth Population in Zhejiang Province from 2014 to 2018

From 2014 to 2018, the number of births from the Statistical Yearbook in Zhejiang Province were 578,000, 581,000, 624,000, 670,000, and 628,000, respectively. The surveillance data were 416,941, 41,490, 434,163, 546,816, and 45,964, respectively. The proportion of surveillance data in the yearbook data was 72.14%, 71.34%, 69.58%, 81.61%, and 71.81%, respectively.

The Statistical Yearbook and surveillance data showed a similar trend. The number of births in 2014 and 2015 were at the same low level, increased in 2016, and reached a peak in 2017, before falling back to a little higher than the 2016 level in 2018. The birth peak of the surveillance data in 2017 was more obvious when compared with the data in 2014, where the Yearbook and surveillance data increased by 15.92% and 31.15%, respectively, as shown in Figure 1.

### 3.2. Impact of the Universal Two-Child Policy on Workload of Maternal Health Management

The maternal health management indicators (early pregnancy registration rate and postpartum visit rate) stabilized at 96.22% to 98.08% in Zhejiang Province from 2014 to 2018. The number of registered early pregnancies and postpartum visits increased by 30.50% and 30.74% in 2017, respectively, when compared with 2014. In 2018, the number of registered early pregnancies dropped to a level slightly higher than that in 2016, which was highly consistent with the trend in the number of births from the surveillance data.

The workload of maternal health management reached its peak in 2017, increasing by 30.37% over 2014 and falling in 2018, but still increased by 7.34% over 2014, as shown in Table 2 and Figure 2.

### 3.3. Impact of the Universal Two-Child Policy on Child Health Management Workload

Among the indicators of child health management in Zhejiang from 2014 to 2018, the rate of newborn home visits was stable between 97.97% and 98.92%, and the rate of health management for children aged 0–6 fluctuated slightly between 94.17% and 98.43%. The number of newborn home visits increased by 29.89% in 2017 when compared with that in 2014, and dropped to a level slightly higher than that in 2016 in 2018, which was highly consistent with the trend of the number of births in the surveillance data. The number of health-managed children aged 0–6 years increased by 9.89% in 2017 when compared with 2014, and continued to increase by 1.74% in 2018.

The workload of child health management peaked in 2017, 12.70% higher than that in 2014. Although it fell in 2018, it was still 7.73% higher than that in 2014, as shown in Table 2 and Figure 3.

### 3.4. Impact of the Universal Two-Child Policy on Children’s Immunization Workload

From 2014 to 2018, the highest dose of primary vaccinations was significantly higher than the number of births in the same year. A peak appeared in 2017, but only increased by 11.61% when compared with 2014, which was lower than the increase in the number of births (even compared with the data from the Statistical Yearbook). The workload of immunization also peaked in 2017, with a small increase (4.33% higher than that in 2014). In 2018, the workload fell back to 1.19% lower than in 2014, as shown in Table 2 and Figure 4.

## 4. Discussion

After the implementation of the universal two-child policy, the birth peak in Zhejiang occurred in 2017, one year later than the national birth peak. The BPHS surveillance data showed a more obvious birth peak than the Statistical Yearbook data. Possible reasons for this are as follows: due to the release of prenatal demand suppressed by family planning policy, the proportion of women with high-risk pregnancies such as pregnant women older than 35 years or with a cesarean delivery history for their first birth, has increased significantly. This has resulted in continuously increasing incidences of pathological pregnancy, pregnancy complications, and a higher proportion of women with a scarred uterus who are pregnant again [9,24,25,26]; therefore, the willingness to seek prenatal examinations and other maternal health management services is stronger. Studies have shown that families with two children have a higher income [8], a higher awareness and acceptance of BPHS [27], and a higher likelihood of receiving maternal health management services. In addition, to implement the universal two-child policy, national publicity including free access to maternal health management has strengthened [28], which helps more pregnant women learn about and accept relevant services.

When compared with the number of births in the Statistical Yearbook, about 20–30% of newborns were not covered by the BPHS from 2014 to 2018. However, during the peak birth rate in 2017, the coverage rate (81.61%) was significantly higher than that in other years (69.58–72.14%). The indicators of maternal and child health management and children’s immunization services remained stable, and the related services did not decline, even in 2017. The number of births in the surveillance data was highly consistent with the number of early pregnancies, prenatal examinations, and newborn visits. This shows that as long as the newborns were covered by CHS, there was a high probability of the newborns and their mothers receiving systematic and continuous BPHS services.

We calculated the workload of related BPHS services (maternal and child health management and children’s immunization) using the equivalent method, and analyzed the workload changes in Zhejiang Province from 2014 to 2018. The data showed that the workload of the related three BPHS services in 2017 (the peak births year) increased by 10.79% when compared with that in 2014 (the universal two child policy had not yet been issued). The greatest impact was observed on maternal health management, which increased by 30.37% in 2017, and the workload of child health management and children’s immunization increased by 12.70% and 4.33%, respectively. Regarding children’s immunization, the increase in single primary vaccine administration was lower than that of births, even when compared with the data from the Statistical Yearbook. The possible reasons for this are as follows: the target children of immunization services in Zhejiang are those who have lived in Zhejiang for over three months, which covers more children than other BPHS services (target population lived there for over six months). One routine task involves conducting quarterly supplementary immunization to catch the children who have missed vaccinations, so the immunization service covers more migrant children than other BPHS services [29]. Primary vaccination targets not only include local and mobile newborns, but also any migrants aged over one year who missed their primary vaccinations. It also explains why the highest dose of vaccines in primary vaccination was significantly higher than the number of births in the Statistical Yearbook of the same year.

When the birth population fell in 2018, the maternal health management workload dropped simultaneously to 82.33% of the level of the previous year. Regarding child health management and children’s immunization, the birth cohort in 2017 was still in the service age groups, so the decline was relatively small. If the number of births continues to decline in the next few years, the maternal health management workload will quickly fall back to an even lower level than the pre-universal-two-child era. However, the child health management workload will decline slowly for about three years, due to the higher EV of the health management of children aged 0–2 years than children aged 3–6 years. The factors influencing children’s immunization workload are complex, except for the change in birth population, which is also affected by the size of the migrant population. In recent years, Zhejiang has continually attracted many immigrants, and the immunization workload will continue to be high.

## 5. Conclusions

The birth population of Zhejiang Province in China peaked in 2017 after the issuance of the universal two-child policy. The indicators of maternal and child health management and children’s immunization services remained stable, and the related services did not decline, even in 2017. The implementation of the universal two-child policy has placed considerable but temporary pressure on maternal health care services, whereas the pressure on child health care services has been lower, but lasts longer. The factors influencing children’s immunization are complex, and the direct impact of the universal two-child policy is relatively minimal.

## 6. Limitations

The limitation of this study was measuring the workload using limited surveillance data, especially given the lack of data on high-risk pregnancy women, high-risk children, and children with nutritional diseases. After the implementation of the universal two-child policy, the proportion of high-risk pregnancies increased significantly as above-mentioned, which also led to a corresponding increase in the incidence of high-risk infants such as premature births [30], coupled with the improvement in screening technology [31]. We did not include relevant work, and therefore underestimated the growth in the maternal and child health management workload.

## Figures and Tables

**Figure 1 ijerph-16-02880-f001:**
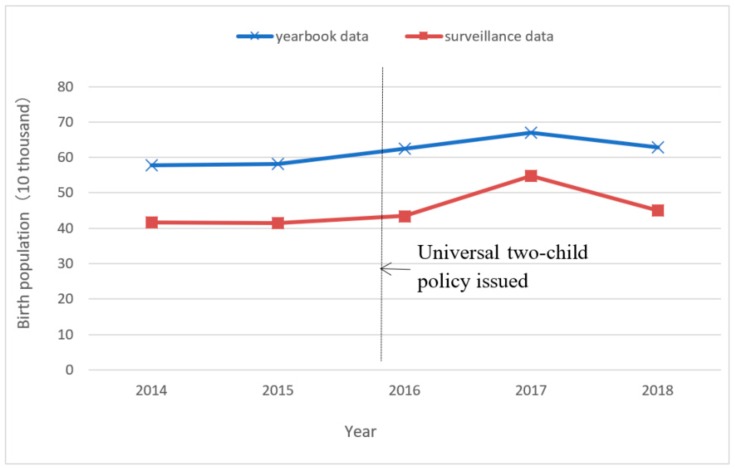
Comparison of the birth population data from different sources in Zhejiang Province, China, 2014–2018.

**Figure 2 ijerph-16-02880-f002:**
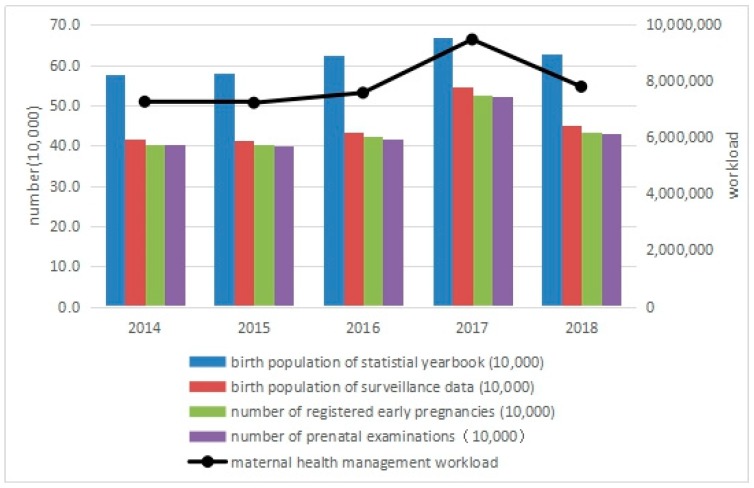
Maternal health management workload in Zhejiang Province, 2014–2018.

**Figure 3 ijerph-16-02880-f003:**
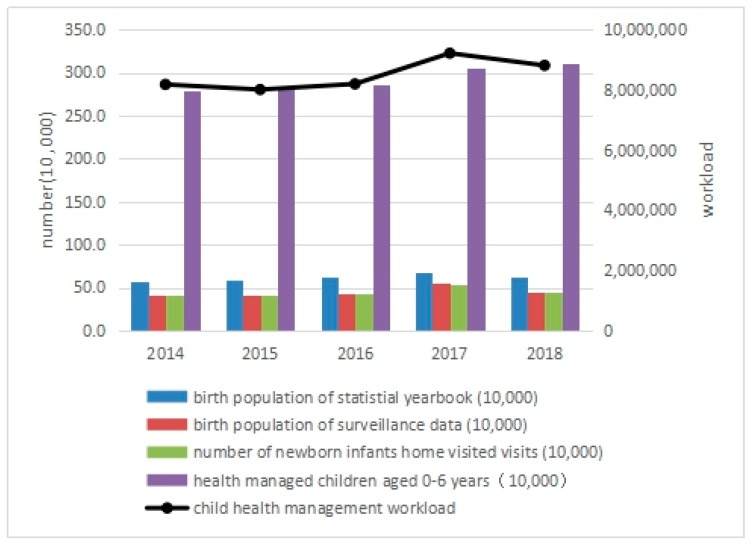
Child health management workload in Zhejiang Province, 2014–2018.

**Figure 4 ijerph-16-02880-f004:**
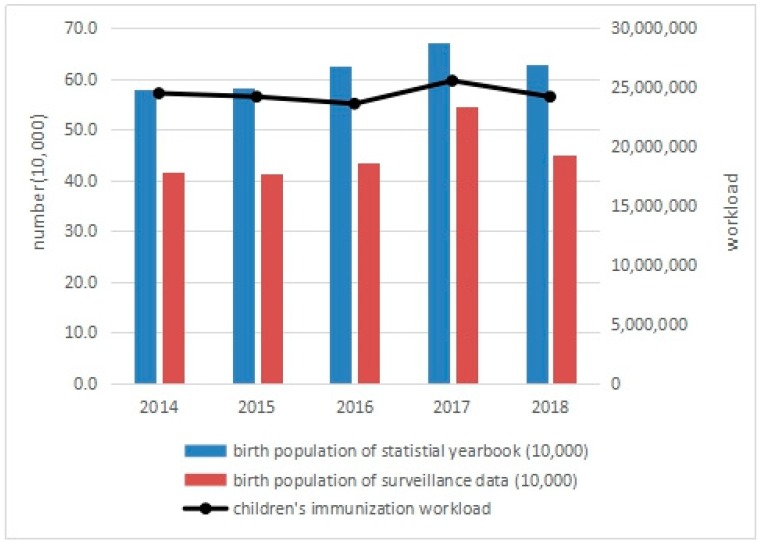
Children’s immunization workload in Zhejiang Province, 2014–2018.

**Table 1 ijerph-16-02880-t001:** The equivalent values (EVs) of related community basic public health services (BPHS) in Zhejiang Province, China.

Category	Sub-Item	EV
Maternal health management	Registration in early pregnancy	8/person
	Prenatal examination (average 4 times per person)	1.5/person–time
	Postpartum visit	3/person
	42-day postpartum examination	1/person
	High-risk pregnancy management	4/person
Child health management	Home visit to newborn infant	3/person
	Health record establishment of newborn infant	0.5/person
	Health management of children aged 0 to 2 years	2.5/person–time
	Health management of children aged 3 to 6 years	2/person
	Health management of high-risk infants and children with nutritional diseases	2/person
Children’s Immunization	Establishment of vaccination certificate	0.5/person
	Vaccination	1.5/dose

**Table 2 ijerph-16-02880-t002:** The workload of related community BPHS services in Zhejiang Province, 2014–2018.

Categories	Indicator	2014	2015	2016	2017	2018
Maternal health management	Number of registered early pregnancies (persons)	403,979	402,569	422,091	527,202	433,913
	Early pregnancy registration rate (%)	96.89	97.12	97.22	96.41	96.22
	Number of prenatal examinations (persons)	402,011	399,066	418,175	522,407	429,830
	Prenatal examination rate (%)	96.42	96.28	96.32	95.54	95.31
	Number of postpartum visits (persons)	407,483	404,712	425,833	532,726	439,277
	Postpartum visit rate (%)	97.73	97.64	98.08	97.42	97.41
	Workload	7,273,830	7,233,796	7,589,110	9,482,962	7,807,392
	Workload change compared with 2014 (%)	-	99.45	104.33	130.37	107.34
	Workload change compared with last year (%)	-	99.45	104.91	124.95	82.33
Child health management	Number of newborn infants home visited (persons)	412,422	409,552	428,295	535,690	441,896
	Home visit rate of newborn infant (%)	98.92	98.81	98.65	97.97	97.99
	Children aged 0–6 years (persons)	2,954,708	2,877,405	2,919,699	3,135,183	3,189,609
	Health managed Children aged 0–6 years (persons)	2,782,514	2,832,300	2,865,019	3,046,490	3,099,491
	Health management rate of Children aged 0–6 years (%)	94.17	98.43	98.13	97.17	97.17
	Estimation value of children aged 1–2 years (persons)	818,150	811,316	821,974	837,847	963,985
	Workload	8,177,737	8,007,346	8,195,021	9,216,661	8,809,646
	Workload change compared with 2014 (%)	-	97.92	100.21	112.70	107.73
	Workload change compared with previous year (%)	-	97.92	102.34	112.47	95.58
Children’s Immunization	The highest dose of vaccines in primary vaccination (persons)	770,832	811,508	716,266	860,365	831,565
	Total doses (no.)	16,063,652	15,857,333	15,492,062	16,740,286	15,849,757
	Workload	24,480,894	24,191,754	23,596,226	25,540,612	24,190,418
	Workload change compared with 2014 (%)	-	98.82	96.39	104.33	98.81
	Workload change compared with previous year (%)	-	98.82	97.54	108.24	94.71
Total	Workload	39,932,461	39,432,896	39,380,357	44,240,235	40,807,456
	Workload change compared with 2014 (%)	-	98.75	98.62	110.79	102.19
	Workload change compared with previous year (%)	-	98.75	99.87	112.34	92.24

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
