# Peer review of "Impact of the Universal Two-Child Policy on the Workload of Community-Based Basic Public Health Services in Zhejiang Province, China"

_ijerph, 2019, doi:10.3390/ijerph16162880_

Round 1

Reviewer 1 Report

This is a really interesting manuscript that explores the outcomes of the changing child policy in China.  The authors have a very important data set here, and are moving in the right direction to explore this resource.  However, a few items that I would like to share:

-Some typos and grammar, but overall very good.

-I am not sure about the equations to calculate the measures.  More details on these complex equation would be good, and would provide a platform to more fully understand the data.  Are these standard within the CDC, were they developed for this study, have they been validated?  Many questions that should be rather easy to address.

-last point, was there any ethics review requirements on this secondary analysis?  A sentence or 2 would have been good (unless I have missed it in the mansucirpt)

-Proportions were calculated, but were any inferential statistics conducted on these data?  At the moment these are trend lines, which is good, but further exploration should provide greater impact of the data.

-It seems to me that the data are really about the degree to which people accessed services for which they were eligible (vaccination or no vaccination).

-the conclusions are a little obvious in my opinion (ie peak of population growth in a region).  Moreover, attributing the policy to population growth is also a little superficial, and so I am not seeing in the manuscript the impact of these association (if they are indeed associated variables).   This is in contrast to the tremendous data that has been extracted.  I just wonder if there were a way to conduct secondary analysis to try and grasp which factors had influent over which variables. I would also be interested in the seeming outcome that the birth rates are about what they were prior to the policy inflection.  

Reviewer 2 Report

Comments in general:

A broad literature review and comparison of this study with other studies on the impact of two children policy in China is necessary.

There are peak effects in 2017, The greatest impact is on maternal health management, increasing by 30.37% in 2017, and the workload of child health management and children’s immunization increased by 12.70% and 4.33%, respectively. Could you please provide statistical significance test for the differences by years in all the tables and figures?

The interpretation of the results are not very clear, which needs to be carefully checked and improved.

Reviewer 3 Report

This paper could be greatly improved with a professional edit. There are numerous error and awkward wording throughout the manuscript.

I would suggest the authors ponder more on the significance of this topic in the introduction. I am not that compelled this is an important topic as claimed by the author.

The increase in birth after the discontinuation of one child policy in China is very interesting to see. However, I would suggest the authors put more weights on the reasons why there is a decrease in 2017 during the posttreatment periods. More discussions can be added to the discussion part and it would complement the analysis of the birth increase. 

Is it possible to distinguish between rural and urban areas? I am thinking the impact of abandoning the one-child policy should have a larger impact on urban residents. It is interesting to find out.

Round 2

Reviewer 2 Report

see the attached document. 

Reviewer 3 Report

Authors improved the quality of the paper. Good job.

Author Response

Thanks for your positive and encouraging comments on the paper. We appreciate the careful reading of this manuscript.